# A Three-Dimensional Hough Transform-Based Track-Before-Detect Technique for Detecting Extended Targets in Strong Clutter Backgrounds

**DOI:** 10.3390/s19040881

**Published:** 2019-02-20

**Authors:** Bo Yan, Na Xu, Wen-Bo Zhao, Lu-Ping Xu

**Affiliations:** 1School of Aerospace Science and Technology, XIDIAN University, 266 Xinglong Section of Xifeng Road, Xi’an 710126, China; boyan@xidian.edu.cn (B.Y.); wbzhao@xidian.edu.cn (W.-B.Z.); 2School of Life Sciences and Technology, XIDIAN University, 266 Xinglong Section of Xifeng Road, Xi’an 710126, China; nxu_1994@stu.xidian.edu.cn

**Keywords:** extended target tracking, track-before-detect, Hough Transform

## Abstract

Hough Transform (HT), which has a low sensitivity to local faults and good ability in suppressing noise and clutters, usually applies to trajectory detection in a cluttered environment. This paper describes its application for detecting the trajectories of extended targets in three-dimensional measurements, i.e., a two-dimensional positional information and its measuring time. For taking the full merits of a multi-scan, the measuring time is regarded as a variable for the time axis. This correspondence extends the HT to 3-dimensional data. Meanwhile, a three-dimensional accumulator matrix is built for the purpose of voting. The voting process is done in an iterative way by selecting the 3D-line with the most votes and removing the corresponding measurements in each step. The three dimensional Hough Transform-based extended target track-before-detect technique (3DHT-ET-TBD), proposed here, is suitable to track the extended target and non-extended target simultaneously and few false alarm trajectories arise. Both the real data and simulated data are exploited to evaluate its performance. Compared with the Gaussian Mixture Probability Hypothesis Density (GM-PHD) filter based methods and a 4DHT-TBD algorithm, the 3DHT-ET-TBD is a more promising approach for multi-extended target tracking problems due to its high efficiency and low computation, especially in situations where the noise and false alarms are considerably high but few measurements are generated by the extended targets.

## 1. Introduction

The challenge of detecting low radar cross section (RCS) targets in a competing clutter arises in many radar applications. The signal to noise ratio (SNR) is so low that the target could not possibly be detected in a single frame. The best way to enhance the SNR is to apply the track-before-detect (TBD) technique to extract the target information through successive frames [1]. TBD methodology has been employed to integrate the energy of targets through a time sequence of frames for the detection of dim targets. Various TBD algorithms arise for the problem of target tracking recently. The algorithms mainly fall into three categories: a particle filter based track-before-detect (PF-TBD) algorithms [2,3,4], dynamic programming based track-before-detect (DP-TBD) algorithms [5,6,7] and Hough Transform based track-before-detect (HT-TBD) algorithms [1,8,9]. The algorithms in References [2,3,5,6] are designed for the targets in video data and the algorithms in References [4,7] are processing points where each point represents one potential target. 

However, for the increased resolution of modern radars, the high resolution allows the target to be found in several cells and then the radars are able to receive more than one measurement per time step from different corner reflectors of a single target. The target is unsuitable to be categorized as a point. This work considers the multi-extended target tracking problem through the use of the Hough Transform and the proposed approach is called the 3DHT-ET-TBD in this context. The input of the 3DHT-ET-TBD is the measurements during a period of time, each measurement is a 3-dimensional point, i.e., a two-dimensional positional information and its measuring time. TBD approaches for the extended targets based on the particle filter and dynamic programming have been developed in References [4] and [10] respectively. Meanwhile, various HT-TBD algorithms have been developed for the detection [1] and the initiation [11,12] of non-extended targets. Meanwhile, the HT algorithm has been widely used in straight line detection [13]. More recently, the Hough Transform has been used to find the time-of-arrival (TOA) curves of the targets in the scans for impulse-radio UWB multistatic radars [14], the target TOA points can be well discriminated from false alarms and associated overscan time for taking full use of the multi-scan. However, the HT algorithm, to the best of our knowledge, has previously not been used in the multiple extended targets tracking (METT) problem. No such a 3DHT-ET-TBD for the purpose of detection and tracking an unknown number of multiple extended targets, in the presence of missed detection and clutter has ever been developed. 

The most popular and widely used extended target tracking methods [15,16,17] are based on the probability hypothesis density (PHD) [18,19]. The algorithms [15,16,17] are capable of estimating the target extents and measurement rates as well as the kinematic state of the target. The number of measurements generated by a target of interest at each time step usually follows a Poisson distribution [20]. The random matrix framework [21] is adapted into the extended target PHD framework [18]. This type of approach allows the problem of estimating multiple targets in clutter and uncertain associations to be cast in a Bayesian filtering framework. It simultaneously associates all measurements with all tracks, rather than attempting to enumerate and rank a list of possible measurement-to-track associations. Meanwhile, this strategy avoids the computational risks in data association algorithms [22,23,24], i.e., the number of feasible joint events grows drastically with the increase of the number of tracks and the number of measurements. However, an important part of obtaining the closed-form solution to the PHD filters is partitioning the measurement sets [25,26]. Theoretically, the above filters require all possible partitions of the current measurement set for its update. However, the number of possible partitions grows very large as the number of measurements increases. Even only a subset of all possible partitions is considered, it brings huge computation with alternative partitioning. Compared with the PHD based methods, the TBD algorithm which integrates the Hough Transform would bring four merits. 

Firstly, unlike the current extended target PHD (ET-PHD) based filters that use only the data present in the current scan, our 3DHT-ET-TBD uses multiple scans (including the current scan and some past scans). Secondly, the idea of the Hough Transform is to make the infinite space of all possible lines finite by discretization of the parameter space and to let each point “vote” for all lines to which it belongs in this parameter space [27]. Therefore, partitioning the measurements into sets is avoided. Meanwhile, data association is unnecessary here and computational risks are also avoided in a multi-track situation. Thirdly, the measurements of other targets would confuse the correction step of PHD filters when the targets are closely distributed. In 3DHT-ET-TBD, it can be alleviated by multiple scan strategy and voting. Fourthly, extended target and non-extended target can be detected in 3DHT-ET-TBD simultaneously. 

Most of the existing line detection methods are based on the standard Hough Transform (SHT) technique [28]. Usually, an HT method consists of the following phases: Hough transformation, voting, peaks localization, determining the actual parameters and verification [29]. In the 3DHT-ET-TBD, three modifications are made for the multiple extended targets tracking the problem. Firstly, the voting process is done in an iterative way by selecting the trajectory with the most votes and removing the corresponding measurements in each step until no trajectories can be detected. All the trajectories in the surveillance area can be obtained one by one. Secondly, the 3-dimensional Hough Transform technique [9] applied here makes it unnecessary to ensure that the target observations are evenly spaced in time. It is superior to the HT-TBD and ET-PHD, which use the sequence number of frames as the measuring time [1,15,16,17,30,31]. Therefore, it also works well in sector scan radar where the interval of measuring time is unequal.

The remainder of the work is organized as follows. Section 2 defines the models and problems. Section 3 embeds the extended tracking problem into HT-TBD algorithm. In addition, in this section, the detailed description and implementation of 3DHT-ET-TBD is presented. To compare the performance of the existing approaches and the proposed method in detail, Section 4 demonstrates the effectiveness through the Monte Carlo simulation and real data of an air surveillance radar. Section 5 draws conclusions.

## 2. Models

### 2.1. Preliminaries

The extended target state ***ξ**_k_* at *k*th scan is defined as the triple ***ξ****_k_*= (*γ*, ***x****_k_*, ***X****_k_*) [15]. The number of targets at the *k*th scan is represented by a variable *N*^T^*_k_*. Firstly, the random variable *γ* > 0 is the measurement rate that describes how many measurements the target, on average, generates per time scan. It usually assumes that the number of measurements generated by one source (a target or a clutter) at each time step is Poisson distributed [20]. Secondly, ***x****_k_’* = (***p****_k_’*, *v_k_*, *α_k_*)^T^∈ℝ^4^ is the kinematic state. ***p****_k_’* describes the target’s position where ***p****_k_’* = (*x_k_’*, *y_k_’*). *v_k_* denotes the velocity and *α_k_* represents the course of the target. Finally, ***X****_k_* is the extension of the target and it usually describes the size and shape of the target [15]. The spatial distribution of the measurement is related to the parameters of the sensor and the real size of the target. The shape model of an extended target is presented in References [32,33]. The shape is usually assumed to be an ellipse because it is a good combination of an informative shape model and low computational complexity. The dynamic models and sensor measurement processes related to the state of *j*th target ***ξ****_k_^j^* at *k*th scan are given by Equations (1) and (2) respectively.
(1)$$\xi_{k+1}^{j}=F(\xi_{k}^{j},\sigma_{k})$$
(2)$${\left\{z\right\}}_{k}^{j}=H(\xi_{k}^{j},\omega_{k})$$
where *F* (*•*) is the state propagation function and *H* (*•*) is the measurement function. Process noise *σ_k_* and measurement noise *ω_k_* are zero mean, white and uncorrelated Gaussian noise sequences. In Equation (2), {***z***}*_k_^j^* denotes the measurements of the *j*th extended target at the *k*th scan. |{***z***}*_k_^j^*| is the number of elements in {***z***}*_k_^j^*. Then, according to the Poisson distribution [20], it has
(3)$$P(|{\left\{z\right\}}_{k}^{j}|=n|\gamma_{k}^{j})=\frac{{\left(\gamma_{k}^{j}\right)}^{n}}{n!}\exp(\gamma_{k}^{j}).$$

The set of measurements generated by clutter is denoted by {***z***}*_k_*^0^. The cardinality of the set |{***z***}*_k_*^0^| is also assumed as a Poisson distributed random variable with rate *γ_k_*^0^. The likelihood of a single false alarm is assumed homogenous but unknown in this contest. The set of measurements ***Z****_k_* obtained at time *k* is the collection of measurements generated by targets and clutters.
(4)$$Z_{k}={\left\{z\right\}}_{k}^{1}\cup \ldots \cup {\left\{z\right\}}_{k}^{N^{T}{}_{k}}\cup {\left\{z\right\}}_{k}^{0}.$$

The set of all the measurements in a time series is denoted by ***Z****^K^*, ***Z****^K^* = {***Z****_k_*}*_k_*
_= 1_*^K^*. Each measurement ***z****_k_* usually consists of a kinematic (position) measurement component (*x_k_^i^*, *y_k_^i^*) and a time stamp records the received time *t_k_^i^*. However, the source of the measurement is unknown. Therefore, the measurement set ***Z****^k^* can also be represented by
(5)$$Z^{k}=\{z|i=1,\ldots ,N_{k}\}=\{(x_{}^{i},y_{}^{i},t_{}^{i})|i=1,\ldots ,N_{k}\}.$$

The input of our 3DHT-ET-TBD is the measurement set ***Z****^k^*. 

### 2.2. Problem Statement 

The TBD algorithm is a method to improve the detection of weak targets by integrating their signal returns over multiple consecutive scans, i.e., estimate the state of targets in each scan ***ξ*** by measurement ***Z****^k^*.
(6)$$\xi =\{\xi_{k}{\}}_{k=1}^{K},\xi_{k}={\left\{\xi_{k}^{j}\right\}}_{j=1}^{N_{k}^{T}}$$

The optimal estimation, which has a maximum likelihood is:(7)$$\xi =\arg\max P(\xi |Z^{k}).$$

In the ET-PHD filters, the measurements of each scan are clustered into sets, each set is the measurements generated by the same source, every set of every partition are utilized to update the state of each target. Considering all possible partitions, ET-PHD has a quadratic growth in computation with a linear increase in the number of measurements. Simplification in the algorithm carries a decrease in performance inevitably. 

In essence, we can fit potential trajectories in the original measurement space directly and implement the optimal multi-target data association strategy by enumerating and evaluating all feasible joint measurement to track associations. However, unlike the most recent work in Reference [34], the trajectories in this work are unknown. This optimal strategy carries computational risks, as the number of feasible joint events grows combinatorially with the number of tracks and the number of measurements. Thus, this approach becomes impractical for a relatively small number of extended targets.

However, HT has a linear growth with a linear increase in measurements. HT utilizes the fact that each of the points in the multiple scans generated by one target lies on a straight line. The HT is the estimation of the parameters of the straight line by voting of these points. Voting enables us to classify these points in the original measurements by accumulating the votes and detecting the peaks in the accumulator space [35]. The detection of the peaks in the accumulator space transforms the parameter estimation problem to a search problem.

## 3. Hough Transform Based TBD Algorithm for Extended Targets 

### 3.1. Introduction of the 3DHT-ET-TBD 

In a narrow sense, the term Hough Transform refers to building an accumulator array with each cell representing one set of parameters from the discretized parameter space, and to increase for each point the counter of all cells to which this point might belong [27]. The resulting accumulator array can thus be considered a transform of the original point into the parameter space. This Hough Transform is the essential subroutine in the proposed 3DHT-ET-TBD. The input of 3DHT-ET-TBD is the 3D measurements during a period of time, such as the measurements of several successive frames. The outputs are the 3D-lines, which consist of 3D-points where each line regards a trajectory and the points in the line means the measurements generated by the same target. 

### 3.2. Parameter Space Discretization 

The parameter space discretization is the most fundamental constituent of this Hough Transform. The 3D-line representation is usually in the vector form of ***a*** + ***b***, where ***a*** = (*x*, *y*, *t*) is a point on the 3D-line and ***b*** is the direction of the line. The direction of a line can be specified by (*φ*, *ϕ*), where *φ* is for horizontal orientation (azimuth), and *ϕ* is for altitude (elevation). When the position of the 3D-line is represented by an arbitrary anchor point ***a*** this leads to three parameters, one of which is redundant. The optimal line representation [36] is one way to remove this redundancy. Roberts first defined a plane which passes through the origin and is perpendicular to the line. The two parameters $\left(x^{\prime },y^{\prime }\right)$ are then defined as the coordinates of the intersection of the line and the plane in the plane’s own 2D coordinate frame of the plane. From an arbitrary point ***a*** on the line, the parameters $\left(x^{\prime },y^{\prime }\right)$ are obtained with
(8)$$\{\begin{array}{c} x^{\prime }=(1-\frac{e_{x}{}^{2}}{1+e_{t}})x_{i}-(\frac{e_{x}e_{y}}{1+e_{t}})y_{i}-e_{x}t_{i}\\ y^{\prime }=-(\frac{e_{x}e_{y}}{1+e_{t}})x_{i}+(1-\frac{e_{y}{}^{2}}{1+e_{t}})y_{i}-e_{y}t_{i} \end{array}$$
(9)$$b=\left(\begin{array}{l} e_{x}\\ e_{y}\\ e_{t} \end{array}\right)=\left(\begin{array}{c} \cos\theta \cos\phi \\ \sin\theta \cos\phi \\ \sin\phi \end{array}\right).$$

At this point, the arbitrary lines can be defined by a vector which consists of four parameters, i.e., $\left(\phi ,\theta ,x^{\prime },y^{\prime }\right)$. Then, the line direction ***b*** is discretized for further simplifying the parameter vector. The tessellation of Platonic solids in Reference [4] is utilized. There are only a finite number of Platonic solids, and the solid with the highest number of vertices is the icosahedron. To obtain more than the 20 icosahedron vertices, each triangle on the surface of the icosahedron can be divided into four new ones using polygon triangulation by inserting a new vertex between each pair of adjacent vertices of the triangle and normalizing its length [27]. The algorithm terminates after conducting the three icosahedron subdivision steps, which leads to 1280 different direction vectors. However, not all the direction vectors are suitable to indicate the direction of a trajectory. In this work, the line direction is corresponding to the kinematic state of the target. *θ* represents the course of the target in 2D coordinate plane and *ϕ* means the velocity. The specific relationship of the variates can be concluded by (10)
(10)$$\{\begin{array}{l} a=\frac{e_{y}}{e_{x}}=\tan(\theta )\\ v=\frac{\sqrt{e_{x}^{2}+e_{y}^{2}}}{e_{t}}=\arctan(\phi ) \end{array}.$$

For the limitation of the target velocity, *ϕ* should meet the following criterion:(11)$$\{\begin{array}{l} v_{\min}\leq \arctan(\phi )\leq v_{\max}\\ 0\leq \phi \leq \frac{\pi }{2} \end{array}.$$

The *v*_min_ and *v*_max_ are the minimum and maximum velocity of targets and are set as 100 m/s and 1000 m/s respectively. The appropriate direction vectors are presented in Figure 1 in green. Meanwhile, the direction vectors whose corresponding velocity larger than the up limit *v*_max_ and less than the lower limit *v*_min_ are indicated by the points and triangles in red and blue colors respectively. The appropriate direction vectors are devoted to its application for making a list where each appropriate direction vector is assigned a number. Then, the direction of the line can be indicated by one parameter, i.e., the sequence number of the direction in the list. The quantity of the direction vectors is represented by *N_d_* here. 

It is worth noting that first translate the measurements such that the center of its bounding box coincides with the origin is essential in the discretization of the $\left(x^{\prime },y^{\prime }\right)$ plane. Otherwise, the Hough space would become unnecessarily large with many cells representing lines outside the bounding box. This centering is done by translating each point as follows
(12)$$\left(\begin{array}{c} x_{}^{i}\\ y_{}^{i}\\ t_{}^{i} \end{array}\right)\leftarrow \left(\begin{array}{c} x_{}^{i}\\ y_{}^{i}\\ t_{}^{i} \end{array}\right)-\frac{1}{2}\left(\begin{array}{c} \max(\{x_{}^{i}|i=1\ldots N_{k}\})-\min(\{x_{}^{i}|i=1\ldots N_{k}\})\\ \max(\{y_{}^{i}|i=1\ldots N_{k}\})-\min(\{y_{}^{i}|i=1\ldots N_{k}\})\\ \max(\{t_{}^{i}|i=1\ldots N_{k}\})-\min(\{t_{}^{i}|i=1\ldots N_{k}\}) \end{array}\right).$$

The $\left(x^{\prime },y^{\prime }\right)$ plane can be divided into *N_x_* × *N_y_* grid cells and the two parameters are utilized to indicate the arbitrary grid cell. Since three parameters are enough to indicate a line, the parameter space of the HT has 3 dimensions, i.e., the parameter space is an accumulator array which has *N_x_* × *N_y_* × *N_d_* number of grid cells. 

The next step is to transform the measurements into the parameter space. Each measurement votes *N_d_* times for the accumulator array, i.e., each direction vector has a vote. A location $\left(x^{\prime },y^{\prime }\right)$ can be obtained given the measurement (*x_i_*, *y_i_*, *t_i_*) and a direction vector (*e_x_*, *e_y_*, *e_t_*) illustrated in Equation (9). It assumes that the sequence number of the direction vector (*e_x_*, *e_y_*, *e_t_*) is *n_d_*. The cell (*n_x_*, *n_y_*, *n_d_*) which would be voted by the measurement (*x_i_*, *y_i_*, *t_i_*) can be obtained by:(13)$$\{\begin{array}{c} n_{x}=\left\lfloor (x^{\prime }+\frac{d_{\max}}{2})\frac{d_{\max}}{N_{x}-1}\right\rfloor \\ n_{y}=\left\lfloor (y^{\prime }+\frac{d_{\max}}{2})\frac{d_{\max}}{N_{y}-1}\right\rfloor \end{array}.$$

Parameter *d*_max_ in Equation (13) indicates the maximum distance between the points.
(14)$$d_{\max}=\sqrt{{\left(x_{\max}-x_{\min}\right)}^{2}+{\left(y_{\max}-y_{\min}\right)}^{2}+{\left(t_{\max}-t_{\min}\right)}^{2}}.$$
(15)$$\begin{array}{l} x_{\max}=\max\{x_{}^{i}|i=1\ldots N_{k}\};x_{\min}=\min\{x_{}^{i}|i=1\ldots N_{k}\}\\ y_{\max}=\max\{y_{}^{i}|i=1\ldots N_{k}\};y_{\min}=\min\{y_{}^{i}|i=1\ldots N_{k}\}\\ t_{\max}=\max\{t_{}^{i}|i=1\ldots N_{k}\};t_{\min}=\min\{t_{}^{i}|i=1\ldots N_{k}\} \end{array}$$

### 3.3. Iterative Line Detection and Post-Processing 

So far, we have discussed the 3D Hough Transform which provides a method for transforming the measurements into an accumulator array in the parameter space. Then, the method to find the actual trajectories in this accumulator array would be illustrated in detail. 

Firstly, we are only looking for the highest voted cell. Then, we have back-traced the 3D-line corresponding to the highest voted cell. The measurements belonging to the 3D-line are then identified and removed from the measurement set ***Z****^k^*. Then, the Hough Transform is applied again to the remaining measurements for finding the next trajectory. The iteration stops when all the trajectories have been detected. The proposed 3DHT-ET-TBD has six steps as follows: 

Step 1, we discrete the parameter space for all 3D-lines crossing the measurements. The *N_d_* appropriate direction vectors and an accumulator array, which has *N_x_* × *N_y_* × *N_d_* number of voting cells are defined by the theory in Section 3.2.

Step 2, we perform the 3D Hough Transform of the measurement
set ***Z****^k^* based on the discretization result of step 1. The votes of the measurements are calculated by Equation (8) and Equation (13). An accumulator array which contains *N_d_* × *N_k_* votes can be obtained. 

Step 3, we exploit the 3D-line parameters corresponding to the highest voted accumulator cell. Each voting cell regards a candidate trajectory. First, the candidate trajectory having the highest votes should be identified. Then, the direction of the candidate trajectory can be obtained from the list of direction vectors. 

Step 4, all the measurements close to the candidate trajectory would be found in this step. The criterion for determining whether a measurement ***z*** belongs to a trajectory given in the form of the vector ***a*** + ***b*** is that its perpendicular distance to the trajectory *d* is less than a fixed width threshold *T_d_* in the $\left(x^{\prime },y^{\prime }\right)$ plane of the Hough space.
(16)$$T_{d}<d,d=\left\|z-(a+\langle b,z-a\rangle b)\right\|.$$

The fixed width *d* is proportional to the measurement noise and extension of the target. A larger value of *d* means a higher probability that a point generated by the extended target is associated with its trajectory, i.e., the 3D-line. The probability can be theoretically expressed by:(17)$$N(-d\leq x\leq d|0,\sigma )=\int_{-d}^{d}\frac{1}{\sigma \sqrt{2\pi }}\exp\frac{1}{2}{\left(\frac{x}{\sigma }\right)}^{2}dx.$$

Meanwhile, a larger value of *d* also means a higher probability that a false alarm is associated with the 3D line. The probability is the product of false alarm rate and the circular gate whose radius equals *d*, i.e.,$\gamma^{0}\pi d^{2}$. Therefore, a moderate value of *d* should be selected. 

Step 5, the optimal 3D line should be determined. A minimum vote count *T_v_* is set as a detection threshold of the target trajectories. If the quantity of measurements belonging to a 3D-line is larger than the minimum vote count, the measurements are assumed to have originated from an extended target. The set of these measurements is represented by ***Zt***. The number of the points generated by a target at each time step follows a Poisson distributed whose expectation equals *γ*. Then, the probability of that more than *T_v_* points are generated by the extended target in *w* scans and can be evaluated by:(18)$$\sum_{i=T_{v}}^{\infty }P(i=n|w\gamma )=1-\sum_{i=1}^{T_{v}-1}P(i=n|w\gamma )=1-\sum_{i=1}^{T_{v}-1}\frac{{\left(w\gamma \right)}^{n}}{n!}\exp(w\gamma ).$$

Then, turn to step 6 with the measurements in set ***Zt***. Otherwise, the candidate trajectory contains too few measurements and the measurements remained in the accumulator matrix are regard been originated from clutter.

Step 6, finally we need to identify all the measurements from ***Z****^k^* close to the fitted line. The newly discovered measurements of this trajectory are also put into the set ***Zt***. Then, the 3D line or the trajectory of a target can be represented by the measurement set ***Zt***.
(19)$$Zt=\{x_{i},y_{i},t_{i}|i=1,\ldots ,N\}.$$

Meanwhile, remove the measurements in ***Zt*** from ***Z****^k^* and their votes in the accumulator array. Otherwise, the same trajectory would be obtained in the next iteration. Then, an orthogonal least squares fitting is applied on the set ***Zt*** for a more accurate trajectory. The 3D-line can also be defined by the following expression.
(20)$$\left[\begin{array}{cc} m & a\\ n & b \end{array}\right]\left[\begin{array}{c} t\\ 1 \end{array}\right]=\left[\begin{array}{c} x\\ y \end{array}\right].$$

The problem turns to the calculation of the parameter matrix under the least square criterion. Equation (21) can be obtained when we plug measurements in Equation (17) into Equation (18).
(21)$$\left[\begin{array}{cc} m & a\\ n & b \end{array}\right]\left[\begin{array}{c} t_{1}\ldots t_{N_{k}}\\ 1\ldots 1 \end{array}\right]=\left[\begin{array}{c} x_{1}\ldots x_{N_{k}}\\ y_{1}\ldots y_{N_{k}} \end{array}\right].$$

Equation (21) can be transformed into Equation (22) by multiplying a matrix.
(22)$$\left[\begin{array}{cc} m & a\\ n & b \end{array}\right]\left[\begin{array}{c} t_{1}\ldots t_{N_{k}}\\ 1\ldots 1 \end{array}\right]\left[\begin{array}{cc} \begin{array}{c} t_{1}\\ \vdots \\ t_{N_{k}} \end{array} & \begin{array}{c} 1\\ \vdots \\ 1 \end{array} \end{array}\right]=\left[\begin{array}{c} x_{1}\ldots x_{N_{k}}\\ y_{1}\ldots y_{N_{k}} \end{array}\right]\left[\begin{array}{cc} \begin{array}{c} t_{1}\\ \vdots \\ t_{N_{k}} \end{array} & \begin{array}{c} 1\\ \vdots \\ 1 \end{array} \end{array}\right].$$

The parameters of the 3D-line can be obtained after rewriting Equation (20). All the variables on the right side of Equation (23) are available in Equation (19).
(23)$$\left[\begin{array}{cc} m & a\\ n & b \end{array}\right]=\left[\begin{array}{cc} \sum_{i=1}^{N_{k}}x_{i}t_{i} & \sum_{i=1}^{N_{k}}x_{i}\\ \sum_{i=1}^{N_{k}}y_{i}t_{i} & \sum_{i=1}^{k}y_{i} \end{array}\right]{\left[\begin{array}{cc} \sum_{i=1}^{N_{k}}t_{i}^{2} & \sum_{i=1}^{N_{k}}t_{i}\\ \sum_{i=1}^{N_{k}}t_{i} & N_{k} \end{array}\right]}^{-1}.$$

Then, the location of the target at time {*t_i_*|*I* = 1,…,*N*} can be obtained by Equation (20). The trajectory consists of the location at different times is more accurate than the measurements in Equation (19). Repetition of steps 3 to 6 until the candidate trajectory contains a sufficiently small number of measurements. The trajectories obtained in step 6 are the final result of 3DHT-ET-TBD. 

It should be noted that the 6 steps of the 3DHT-ET-TBD can be divided into two stages. Stage 1 containing steps 1–5 is designed to obtain the optimal track associations, i.e., the points set ***Zt***, by the 3DHT algorithm. Stage 2, containing step 6, is to implement a smoothing method. The position of extended targets at a given time can be improved significantly in accuracy by smoothing, which uses more measurements beyond the current estimation time. The least squares fitting in step 6 is an example to implement the trajectory smoothing. Instead of using the target-state transition model in Equation (1), the state of extended targets in multiple scans is estimated by the points in the set simultaneously. More powerful techniques for this issue [34,37] are available when the trajectory or the measurements of the target is known. It can be expected that more accurate positions can be achieved when the smoothing methods [34,37] are applied in step 6. However, this work is mainly aimed at evaluating the effectiveness of the 3DHT. Therefore, the least squares fitting is utilized here.

For a better description, the roadmap of the steps mentioned above is given in Figure 2a. Meanwhile, the application of the 3DHT-ET-TBD method in METT is presented in Figure 2b. The input of the 3DHT-ET-TBD is the points in several successive scans. A sliding window containing the points of the three scans is presented in Figure 2b. ***T****_k_* is the set of trajectories in sliding window *W_k_*. 

## 4. Simulation Results 

### 4.1. Synthetic Data 

In order to evaluate the performance of the proposed algorithm, 200 Monte Carlo numerical simulations were performed. The simulation was performed on the Intel Core I7-4790, 3.6 GHz with 4 GB RAM in a matlab R2016a environment. The trajectories of 32 closely targets are presented in Figure 3a. The 32 targets are within the surveillance region in all 20 frames and time information is showcased by color. The targets are far from the others at the beginning. In the middle of the trajectories, the targets are closely distributed. Figure 3b shows the simulated date when the measurement rate, measurement noise and false alarm rate are equal to 2, 50 and 6 × 10^−7^. Red and black points are the measurements that originate from the target and clutter respectively. In this situation, it is hard to find the trajectories with naked eyes. The result of 3DHT-ET-TBD is presented in Figure 3c when the measurements in Figure 3b are taken as input. Some position error may exist, but all 32 trajectories have been well detected. The detected trajectories are represented by the lines in different colors and the balls on the line mean the target position in each scan. Meanwhile, no false alarm trajectories arise in Figure 3c. In fact, it is hard to form a false alarm trajectory when we take full use of multi-scan. 

In this work, 10 scenarios are considered to validate both the accuracy and robustness of the algorithms, the measurement rate of targets, the measurement noise and the false alarm rate are various in each scenario. The detailed parameters of the scenarios are presented in Table 1. It is worth noting that the probability of no measurements generated by a target in a scan equals 13.53% and 1.83% when the measurement rate equals 4 and 2 respectively. It is hard to detect or track an extended target if no measurements are generated by it.

The 4DHT-TBD in Reference [14] and Gaussian mixture PHD (ET-GM-PHD) filter [15] is performed to verify the advancement of the 3DHT-ET-PHD. Measurement partition is necessary before the iteration of the ET-GM-PHD filter. Distance partitioning method [15] and K-means++ algorithm [38] are utilized in the PHD filter of [15]. Meanwhile, several partition methods have been developed to promote the ET-GM-PHD filter in performance, including a partitioning algorithm based on affinity propagation (AP) clustering [25] and a method using fuzzy adaptive resonance theory (ART) model [26]. 

The optimal sub-pattern assignment (OSPA) distance is used for evaluating the performance of the algorithms. The OSPA distance between the positions of *n* targets ***T***_1_ = {***T***_1_^1^, ***T***_1_^2^,…,***T***_1_*^n^*} and the positions ***p*** = {***p***_1_, ***p***_2_,…, ***p****_n_*} in each scan can be calculated by:(24)$$\mathrm{OSPA}(T,p)=\{\begin{array}{c} D_{p,c}(T,p),m>n\\ D_{p,c}(p,T),m\leq n \end{array}$$
(25)Dp,c(T,p)=(1n(minκ∈Ω∑i=1m(dc(Ti,pκ(i)))p+(n−m)cp))1p,m≤n

Ω represents the set of permutations of length *m* with elements taken from ***T***. The cut-off value *c* and the distance order *p* of OSPA distance are set as *c* = 150 and *p* = 1 in this work. A smaller OSPA distance can be achieved when the extended targets can be well detected and merely a few false alarm trajectories arise. Figure 4 shows the OSPA distance of the six methods. Figure 4a corresponds to scenario 1, and so on. A smaller OSPA distance means a better tracking performance. 

In scenarios 1 and 6, a very small OSPA distance can be obtained by the integration of the PHD filter and ART partition method for small measurement noise and a low false alarm rate. The cyan lines in scenarios 1–3 and scenarios 1,4,5 respectively showcase that the ART partition method is sensitive to the measurement noise and false alarm rate. 

As to the performance of the integration of the PHD filter and AP partition method, the comparison between scenarios 2,4,5 infers that it works well in a strong false alarm rate. In addition, a similar conclusion can be found in scenarios 7,9,10. However, the results in scenarios 1–3 and scenarios 6–8 infer that the integration of the AP partition method works well only in low measurement noise. A high measurement noise brings a large OSPA distance (scenarios 3 and 8). 

Compared with the above two methods, the integration of the PHD filter and K-means++ the algorithm seems more stable. Reference [15] says that “K-means++ fails to compute informative partitions much too often, except in scenarios with very low false alarm rate”. Therefore, K-means++ algorithm is hard to be used directly in this work for the high false alarm rate. Therefore, a preprocess is added to suppress the false alarm and the number of clusters is set before the K-means algorithm. In this situation, the results in scenarios 1–5 shows that the variation in measurement noise and false alarm rate has not made a drastic fluctuation in OSPA distance and the result is not the most appropriate one. Meanwhile, a large OSPA would arise when the measurement rate of the target is very low (scenarios 6–10). 

In the integration of the PHD filter and distance partition method, it works well with small measurement noise and a low false alarm rate (scenarios 1 and 6). High measurement noise (scenarios 3 and 8) would drastically decrease the tracking performance. It often failed to track the targets when the measurement rate is small and the measurement noise is very large (scenario 8) or the false alarm is very high (scenarios 9–10). Quite a few false trajectories are built for the dense clutter. 

The 4DHT-TBD works well in a high false alarm rate (scenario 5, 10), but drastic deterioration would arise in high measurement noise (scenarios 3 and 8). The performance of the 4DHT-TBD is inferior to that of the 3DHT-ET-TBD in general.

The analysis mentioned above infers that all the methods have their own merits. However, it is hard to detect and track targets if all the undesirable situations come together. However, Figure 4 infers that the proposed algorithm (rad line) is always better than the others under various scenarios. The extended targets can be well detected when the measurement noise and false alarm rate is high. It is better than the others even in scenario 1 where the targets are easy to detect and track. 

The average OSPA distance in different scenarios is presented in Table 2. The lowest OSPA distance in each scenario is emphasized in boldface. It can be seen that 3DHT-ET-TBD is the best almost in all scenarios. The robustness and effectiveness of the 3DHT-ET-TBD under various circumstances have been proved again by Table 2.

Their average running time of the 6 methods is presented in Figure 5. The computation of the method is also significant because a frame of measurements should be processed within a radar scanning cycle. In the PHD based approaches, the computation time consists of two parts, i.e., the time for partition method and time for PHD filter. Those of the PHD based approaches are represented by different color in Figure 5. As a whole, a higher measurement rate (scenarios 1–5) means more computation. The running time of the integration of the PHD filter and distance partition method is large in all 10 scenarios. The computation of K-means++ algorithm is small. However, its corresponding computation in PHD filter is huge. The opposite situation arises in the AP partition method and ART partition method. The whole running time of the two methods is often lower than those of K-means++ algorithm. The running time of 4DHT-TBD is much larger than the others in dense clutter scenarios (scenarios 5,10). It infers that high dimensionality of either the data space or the parameter space results in more computational time and larger memory requirements so that the implementation in engineering may become impractical, especially in dense clutter scenarios.

As to the 3DHT-ET-TBD, it is the most efficient method in scenarios 1–5 and is the second most efficient method in scenarios 6–10. The results infer that the proposed algorithm is computationally efficient. Figure 5 also infers that the computation of the 3DHT-ET-TBD is mainly associated with the number of target measurements. Therefore, the computation of the 3DHT-ET-TBD is usually small and stable. This merit made the 3DHT-ET-TBD being an engineering-friendly method.

### 4.2. Real Data

To evaluate the performance of the proposed algorithm further, we conduct an experiment by an air surveillance radar in a general airport of Pucheng city, ShannXi Provence, P.R.C. Acquisition of the radar was made in January 2016. The real tracks of the targets are obtained by GPS (Global Positioning system) in an airplane. The eight real tracks are presented in Figure 6a and their corresponding measurements are presented in Figure 6b. The total scan number is 45, and the radar scanning cycle is 2 s. Eight targets are within the surveillance region in different periods. Targets 2–4 are always inside the surveillance region, and the target 1,5,6,7,8 enter at 1s and leave at 60 s, 68 s, 60 s, 66 s, 45 s respectively. The measurement rate *γ* of trajectories is different for the difference in the appearance of aircraft. Approximately 4–8 measurements are generated by the target 4 in each scan and that of the target 3 is no larger than 3. Some unexpected misdetection maybe arises in successive scans.

The trajectories obtained by the 3DHT-ET-TBD is presented in Figure 7a. The result is compared with the integration of the PHD filter and different partitioning methods. The OSPA distance of the mentioned methods is presented in Figure 7b. For the benefits of multi-scan, the 3DHT-ET-TBD is more stable than the others. Meanwhile, the OSPA distance of the 3DHT-ET-TBD of all 45 scans is always lower than the others mainly because all the targets are detected and few false alarm trajectories arise. The integration of PHD filter and different partitioning methods always fail to detect targets.

Another weakness of the PHD filter is that plenty of parameters are necessary to be well set. At the beginning of the GM-PHD filter, the real initial state of the extended targets has been told such as the location and velocity. That is the main reason why the PHD filter is superior to the 3DHT-ET-TBD at the beginning. Meanwhile, the parameters in the GM-PHD filter, such as the measurement rate and false alarm rate, have been set to fit the simulated data of each scenario. Table 3 showcases the parameter values in several scenarios. It infers that the initial parameters of the GM-PHD are various in different scenarios.

Meanwhile, the parameters are usually unknown. The measurement rate is not merely related to the parameter of the radar. The Model in Section 2.1 infers that it is also related to the size and course of the target. Therefore, it is hard to find an appropriate measurement rate to fit all the extended targets. In this situation, some misdetection would arise for an inappropriate parameter. All these limitations are harmful to the practicability of the PHD filter. However, in the 3DHT-ET-TBD, merely the minimum vote count and the threshold of distance *d* in Equation (16) are the parameters which are needed to be adjusted in real engineering. The parameter values of the 3DHT-ET-TBD and the 4DHT are presented in Table 4. The parameter values which are utilized in all 10 synthetic scenarios and the real scenario are the same. Fewer parameters allow the 3DHT-ET-TBD more flexibility in usage. Table 4 also infers that the accumulator array of the 3DHT-ET-TBD has *N_x_* × *N_y_* × *N_d_* (100 × 100 × 541) voting cells and that of the 4DHT-TBD has *N_x_* × *N_y_* × *N_v_* × *N_c_* (100 × 100 × 60 × 90) voting cells. The memory requirements of 4DHT-TBD are much larger than that of 3DHT-ET-TBD.

However, a straight-line constant-velocity mobility model is assumed in this work for the target because the HT is designed to extract straight-line target trajectories in the Cartesian data. Therefore, in the HT based TBD algorithms, the performance can be greatly deteriorated if the extended target is maneuvering. Meanwhile, it should be noted that considering the constant-velocity assumption does not necessarily mean that the target must strictly follow this mobility model for the Hough Transform algorithm to work properly. More than one point would be originated from an extended target and the points usually appear as a cloud instead of a single point. This allows more flexibility in the target real movement with respect to the assumed model.

## 5. Conclusions

The 3DHT-ET-TBD is an efficient and powerful method to detect and track the trajectories in various situations, especially in some scenario where both the measurement noise and the false alarm rate are very high. Existing methods which do not take full use of multi-scan fail to detect the extended targets. The advantage of the 3DHT-ET-TBD has been detailed presented in this work by performing real data and simulated data. Meanwhile, some limitations still exist. The trajectory which can be detected by the method is a straight line. Although it is widely accepted that the targets usually move in a straight line at the trajectory initialization stage, we would like to develop more approaches which can be used to detect maneuver extended targets from strong clutter in our later work.

## Figures and Tables

**Figure 1 sensors-19-00881-f001:**
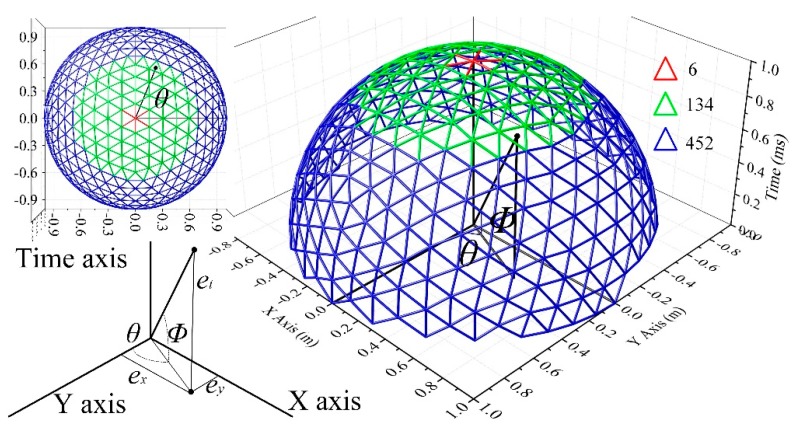
The candidate direction of 3D-lines after three icosahedron subdivision steps.

**Figure 2 sensors-19-00881-f002:**
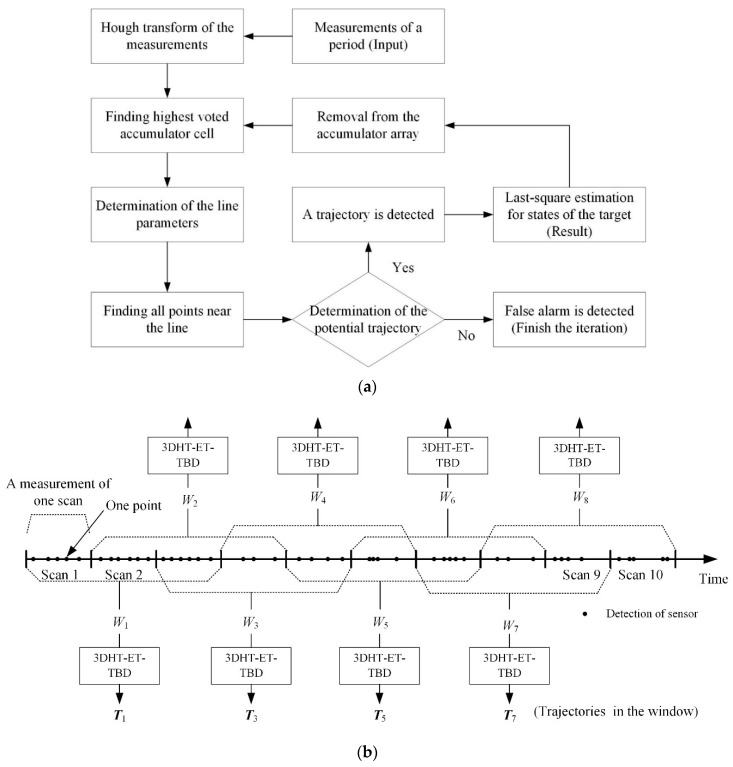
(**a**) The roadmap of the 3DHT-ET-TBD method. (**b**) The application of the 3DHT-ET-TBD method in multiple extended target tracking (METT).

**Figure 3 sensors-19-00881-f003:**
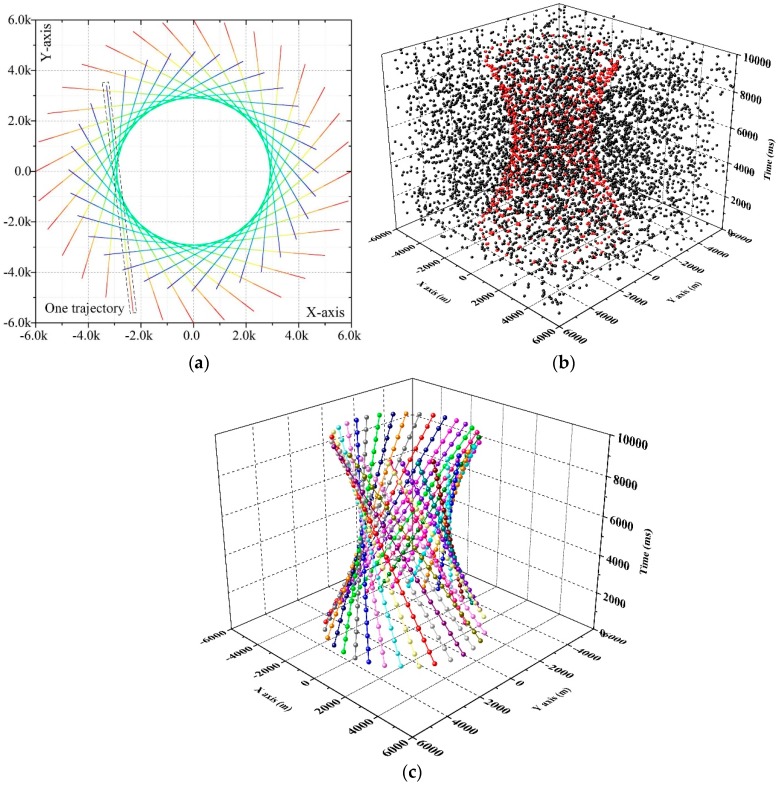
(**a**) The 32 true trajectories in the simulation. (**b**) The simulated measurements of the 32 true trajectories. (**c**) The result of the 3DHT-ET-TBD when the measurements in Figure 3b are taken as the input.

**Figure 4 sensors-19-00881-f004:**
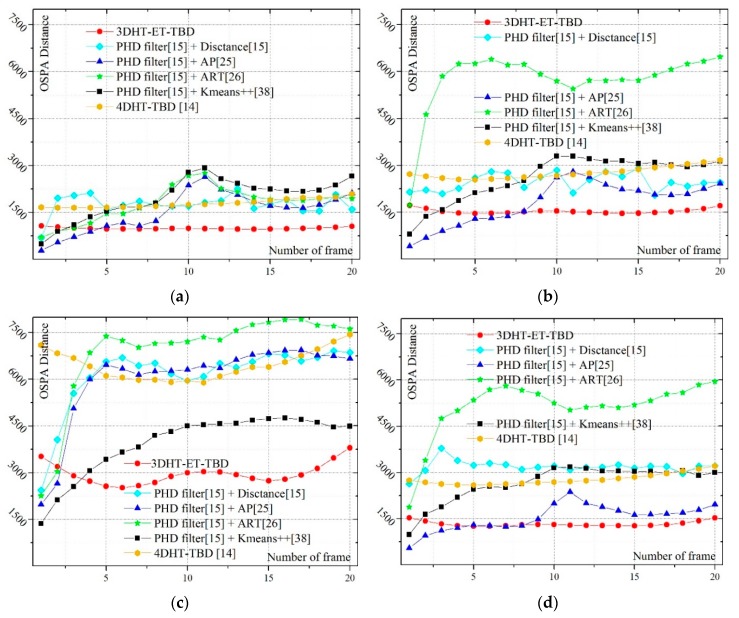
The OSPA distance of 10 scenarios at each scan. (**a**–**j**) corresponds to scenarios 1–10.

**Figure 5 sensors-19-00881-f005:**
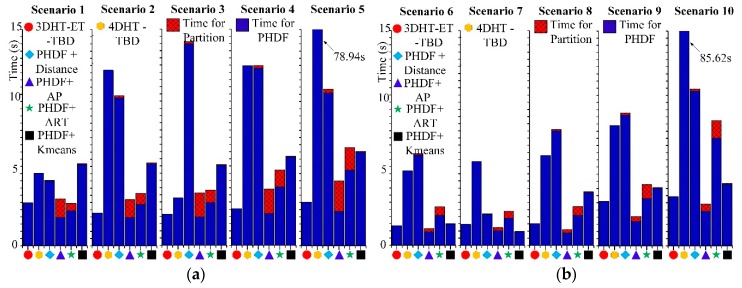
(**a**) The average computation time of scenarios 1–5. (**b**) The average computation time of scenarios 6–10.

**Figure 6 sensors-19-00881-f006:**
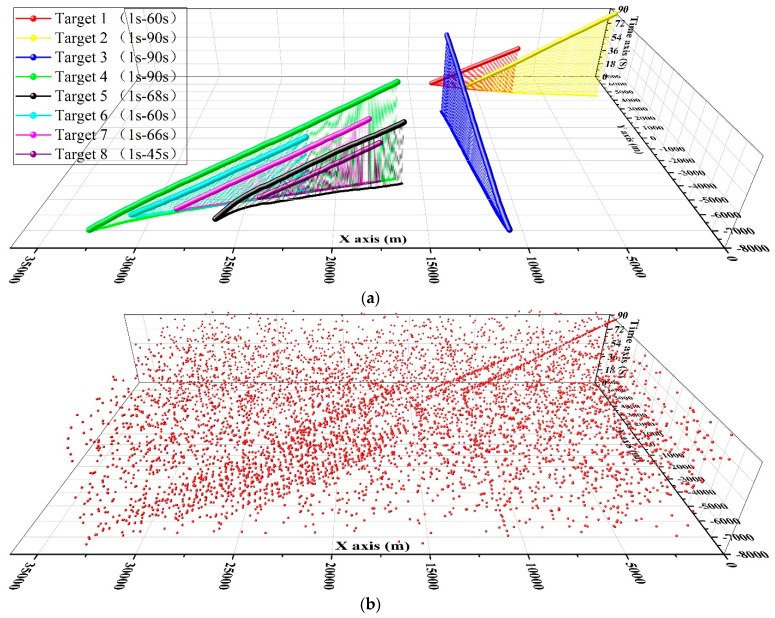
(**a**) Eight real trajectories in the surveillance area. (**b**) The measurements of the eight targets.

**Figure 7 sensors-19-00881-f007:**
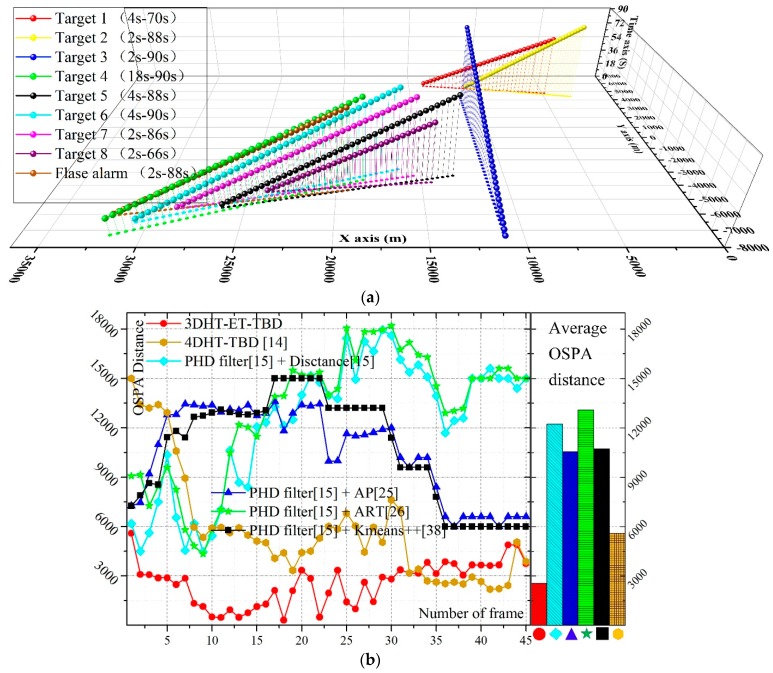
(**a**) The result of the 3DHT-ET-TBD. (**b**) The OSPA distance of the methods at each scan.

**Table 1 sensors-19-00881-t001:** The parameters of the scenarios.

	Measurement Rate γ	Measurement Noise (m)	Number of Clutter Per Square (1/m^2^)
Scenario 1	4	10	6 × 10^−7^
Scenario 2	4	50	6 × 10^−7^
Scenario 3	4	100	6 × 10^−7^
Scenario 4	4	50	1.2 × 10^−6^
Scenario 5	4	50	1.8 × 10^−6^
Scenario 6	2	10	6 × 10^−7^
Scenario 7	2	50	6 × 10^−7^
Scenario 8	2	100	6 × 10^−7^
Scenario 9	2	50	1.2 × 10^−6^
Scenario 10	2	50	1.8 × 10^−6^

**Table 2 sensors-19-00881-t002:** The average the optimal sub-pattern assignment (OSPA) distance of the algorithms in 10 scenarios.

	3DHT-ET-TBD	4DHT-TBD [14]	PHDF [15] + Distance [15]	PHDF [15] + AP [25]	PHDF [15] + ART [26]	PHDF [15] + Kmeans++ [38]
Scenario 1	**993.8**	1791.0	1755.3	1529	1797.6	1976.5
Scenario 2	**1530.8**	2765.4	2467.6	1811.3	5725.1	2604.6
Scenario 3	**2963.1**	6450.5	6132.1	6029	6890.3	3956.7
Scenario 4	**1327**	2782.2	3188.4	1536.3	5108.9	2664.3
Scenario 5	**1600.3**	2820.9	4408.2	1628.5	4468.7	2924.1
Scenario 6	1232.3	2806.1	1540.3	2828.7	**1156.2**	5018.8
Scenario 7	**2152.6**	3747.7	3105.4	3571.2	6212.2	5183.4
Scenario 8	**4303.5**	6634.1	6063.6	4396.7	4665.5	4160
Scenario 9	**2101.1**	3155.8	4874.6	3279.5	3784.3	3017.4
Scenario 10	**2336.5**	3572.5	4861.1	3204.5	3929.1	3499

* The lowest OSPA distance in each scenario is emphasized in boldface.

**Table 3 sensors-19-00881-t003:** Parameter values used for simulations and real data.

	Measurement Rate γ	Probability of Detection and Survival	Covariance of Systematic Error	Covariance of Measuring Error (m,°)	Number of Clutter Per Squre (1/m^2^)
Scenario 1	4	(0.99,0.99)	10	(20, 1.17)	6 × 10^−7^
Scenario 5	4	(0.99,0.99)	50	(20, 1.17)	1.8 × 10^−6^
Scenario 8	2	(0.99,0.99)	100	(20, 1.17)	6 × 10^−7^
Real data	4	(0.99,0.99)	50	(20, 1.17)	6 × 10^−7^

**Table 4 sensors-19-00881-t004:** Parameter values used for simulations and real data.

Parameters in the 3DHT-ET-TBD		Parameter Values Used in the 4DHT	
Number of bins in X axis *N_x_*	100	Number of bins in X axis *N_x_*	100
Width of bins in X axis (m)	160	Width of bins in X axis (m)	160
Number of bins in Y axis *N_y_*	100	Number of bins in Y axis *N_y_*	100
Width of bins in Y axis (m)	160	Width of bins in Y axis (m)	160
Minimum vote count	30	Minimum vote count	30
Threshold of points *d* in Equation (16)	160	Threshold of points *d* in Equation (16)	160
Length of sliding window	7	Length of sliding window	7
Number of bins in 3D direction *N_d_*	541	Number of bins in velocity *N_v_*	60
		Width of bins in velocity (m/s)	15
		Number of bins in course *N_c_*	90
		Width of bins in course (°)	4

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
