# Peer review of "A Three-Dimensional Hough Transform-Based Track-Before-Detect Technique for Detecting Extended Targets in Strong Clutter Backgrounds"

_sensors, 2019, doi:10.3390/s19040881_

Reviewer 1 Report

The present work solves the multi-extended target tracking problem in the challenging 3D space by the means of the Hough transform, which makes use of the TBD measurements of multiple scans. While the literature on the topic of extended target tracking is considerable, the present idea, based on Hough Transform, provides a novel and promising solution. The work is well motivated and makes sense. The simulation study is comprehensive. Overall, the paper has my very positive recommendation.

Some suggestive comments:

1. The Hough Transform has the advantage to make the infinite space of all possible lines finite by a discretization of the parameter space, and then each point votes for the discrete lines to which it belongs in the parameter space. The key idea is to transform the measurements into the parameter space and then do “fitting” in the parameter space. As such, I am wondering whether it is better than directly do fitting (namely detection of the lines) in the original measurement space? For example, the least squares fitting approach is used in the following paper for fitting the measurements for the purpose of tracking a point-target that moves a hybrid of straight lines and smooth curves. This will avoid converting the measurement to the parameter space and converting it back (as well as the discretization of the space). Actually, the orthogonal least squares fitting is mentioned in Step 6. The authors may consider adding a comparison in the simulation or clarification on this issue. This will strengthen the motivation and use of the HT tool (i.e., why not simply the orthogonal least squares fitting?).

Ref. T. Li, Single-road-constrained positioning based on deterministic trajectory geometry, IEEE Communications Letters, Online. 2018

2. An important issue with regard the sliding-time-window fitting (namely multi-scan fitting), as pointed out in the above work (and its reference therein) is to use the fitted parameter at the previous filtering time as the initial parameter to start from for fitting at the current time. This can speed up the nonlinear fitting. While the present work may not directly involve nonlinear fitting, is there also some information about the target trajectories estimated at time k be very useful for the estimation at time k+1? (considering that, the present method considers the data of multiple scans and the considered target trajectories follow smooth curves)

3. It seems one limitation of the HT method is that it only applies to simple curves, mainly straight lines which is also the case of the simulation study. So the authors need to address this limitation explicitly. (In particular, when stating like “So we can say that the 3DHT-TBD is superior to the others in performance and is practical in usage.”, it is better to specify “in what cases”. I don’t believe that one method is better in all cases.)

4. Please clarify whether the proposed method uses the target-state transition model (1). (if not, it is an advantage of the proposed method. If yes, please specify it in the simulation)

5. Please reconsider this sentence in the abstract “ The time is regarded as the third dimension to take full use of multi-scan merits in this Hough Transform.”

6. Please specify “maximizing” over “what” in Eq. (15)? Please check the correct use of the “max” operator. 

7. Several thresholds need to be specified, such as the number of the minimum votes to form a line/trajectory. Please explain each of them in a theoretically solid and explicit way. The parameter d in (16) should also be explained clearer, in more details.

8. There are Chinese characters in Figure 1 and please consider to remove/replace them. 

Language problems: please double check the following sentences, just to name a few:

(1). In lines 145-146, “A points in x-y plane can be represented by a curve in HT domain whose horizontal axis and vertical axis are denotes ρ and θ respectively.”   

(2). In line 172, “When the position of the line is represented by an arbitrary anchor point a, this leads to three parameters, one of which is redundant.”

(3). In line 274, “Repetition of steps 3 to 6 until the candidate trajectory contains too few measurements.”

(4).In line 337, “It mainly because too many false trajectories arise.”

(5) In lines 294-295, “Increasing in detection rate and decreasing in false alarm trajectory make the results are much more appropriate to describe the trajectory of targets.”

(6)In line 417, “K-means++ algorith [17]”

Author Response

The response to the reviewer’s comments ispatched in a file "Response to rewiewer 1.docx".

Reviewer 2 Report

The paper describes three-dimensional hough transformation based track-before-detect technique.

There are following questions to be addressed to make this paper suitable for publication:

- there have been many hough transformation based track-before-detect methods [1, 8-9]; the authors say that 'However, the HT algorithm, to the best of our knowledge, has previously not been used in a framework for tracking an unknown  number of multiple extended targets, in the presence of missed detection and clutter.' Alexiev et al, 2000 A Hough Transform Track Initiation Algorithm for Multiple Passive Sensors claim precisely that. I think it is necessary to contrast those previous findings and outline the novelty of the method. More, the method is only compared with PHD filters which are not very close in terms of the original appeal; is there any way to outline the performance in relation to the more close Hough transform based methods?

- Moreover, it is stated that 'Another weakness of the PHD filter is that plenty of parameters are necessary to be well set... Meanwhile, the parameters in the GM-PHD filter, such as the measurement rate and false alarm rate, have been set to fit the simulated data of each scenario. It means that the initial parameters are various in different scenarios. ' It would be useful to state more precisely how the parameters have been set; have they just been taken to match the simulation parameters? 

- The text requires significant efforts in proofreading; Page 15: 'Globle Positioning System' -> Global Positioning system; P 16: ' some limitation still exist.' -> some limitations still exist; Page 11:  'Turn to the OSPA distance of all the algorithms tested here at each scan.' --- unclear; 'Increasing in detection rate and decreasing in false alarm trajectory make the results are much more appropriate to describe the  trajectory of targets.' --- unclear; P 9: 'Repetition of  steps 3 to 6 until the candidate trajectory contains too few measurements.' -> Repetition of  steps 3 to 6 until the candidate trajectory contains a sufficiently small number of measurements.

- Section 3.2 is devoted to the description of the well-known Hough transform; it would be sufficient to give a reference and contrast instead the novelty of the proposed method against the previously existing ones for tracking-by-detection with Hough transform.

Author Response

Thank you very much for your comments about our paper submitted to Sensors (sensors-427536). We have checked the manuscript and revised it according to the comments. The response to the reviewer's comments is patched in a file "Response to reviewer 2.docx". 

Round  2

Reviewer 2 Report

This revision has been improved significantly, and the authors have sufficiently addressed my concerns.